# Fusarium Head Blight Infection Induced Responses of Six Winter Wheat Varieties in Ascorbate–Glutathione Pathway, Photosynthetic Efficiency and Stress Hormones

**DOI:** 10.3390/plants12213720

**Published:** 2023-10-30

**Authors:** Katarina Sunic, Lidija Brkljacic, Rosemary Vukovic, Zorana Katanic, Branka Salopek-Sondi, Valentina Spanic

**Affiliations:** 1Department for Cereal Breeding and Genetics, Agricultural Institute Osijek, Južno predgrađe 17, 31000 Osijek, Croatia; katarina.sunic@poljinos.hr; 2Ruđer Bošković Institute, Biljenička cesta 54, 10000 Zagreb, Croatia; lidija.brkljacic@irb.hr (L.B.); branka.salopek.sondi@irb.hr (B.S.-S.); 3Department of Biology, Josip Juraj Strossmayer University of Osijek, Cara Hadrijana 8/A, 31000 Osijek, Croatia; rosemary.vukovic@biologija.unios.hr (R.V.); zorana.katanic@biologija.unios.hr (Z.K.)

**Keywords:** FHB, abscisic acid, salicylic acid, AsA-GSH cycle, photosynthetic efficiency, wheat

## Abstract

Fusarium head blight (FHB) is one of the most studied fungal diseases of wheat, causing massive grain yield and quality losses. This study aimed to extend previous studies on the physiological and biochemical responses of winter wheat to FHB stress in a controlled environment by focusing on the ascorbate-glutathione pathway (AsA-GSH), photosynthetic efficiency, and stress hormone levels, thus providing insight into the possible interactions of different defense mechanisms during infection. The activity of AsA-GSH metabolism was increased in FHB resistant varieties, maintaining the redox state of spikes, and consequently preserving functional photosystem II. Furthermore, carotenoids (Car) were shown to be the major pigments in the photosystem assembly, as they decreased in FHB-stressed spikes of resistant and moderately resistant varieties, compared to controls. Car are also the substrate for the synthesis of abscisic acid (ABA), which acts as a fungal effector and its elevated content leads to increased FHB susceptibility in inoculated spikes. The results of this study contributed to the knowledge of FHB resistance mechanisms and can be used to improve the breeding of FHB resistant varieties, which is considered to be the most effective control measure.

## 1. Introduction

About 20% of the world’s annual wheat grain yield and quality losses are due to various plant damages caused by fungal pathogens [1,2]. Among them, *Fusarium* species causing the disease Fusarium head blight (FHB) are listed among the top ten pathogens globally based on their importance [3]. It is also one of the most widespread [2] and the most intensively studied wheat diseases [4]. Although significant progress has been made in the control of FHB, this disease remains a serious threat, especially in terms of accumulation of mycotoxins that are harmful to human and animal health [4,5,6]. The development and intensity of FHB can be conditioned by different factors related to host susceptibility and pathogen aggressiveness, but the most important are weather conditions during anthesis [5,7,8]. Extreme weather conditions associated with climate change are expected to lead to various changes in *Fusarium* species, such as changes in seasonal phenology, population dynamics, and geographic distribution [9], while an increase in FHB epidemics as a result of global warming has already been recorded [10,11]. Effective FHB management requires a combination of cultural, biological, chemical, and host plant resistance strategies, where genetic control accomplished by breeding gains, when combined with other methods, may be a sustainable solution for FHB control [6,12]. However, breeding FHB resistant varieties with desirable agronomic traits is challenging because most sources of FHB resistance correlate with unfavorable agronomic traits [13]. Considering the fact that wheat is exposed to numerous pathogens, it has developed a sophisticated network of mechanisms to escape this threat [14].

Reactive oxygen species (ROS) production is one of the earliest plant responses following pathogen recognition [15,16]. Unstable ROS homeostasis during pathogen attack can lead to oxidative damage and plays a critical role in plant defense responses [16,17,18]. To maintain the balance between ROS scavenging and frequent generation, plants have developed an antioxidant defense machinery comprised of enzymatic and non-enzymatic antioxidants [19]. A crucial role in ROS scavenging by the antioxidant system is assigned to the ascorbate–glutathione cycle (AsA-GSH), which consists of the two main redox buffers of the plant cell, ascorbic acid (AsA) and glutathione (GSH), and the enzymes ascorbate peroxidase (APX), monodehydroascorbate reductase (MDHAR), dehydroascorbate reductase (DHAR), and glutathione reductase (GR) [20]. Apart from its role in ROS scavenging, the AsA-GSH cycle also regulates the signaling potential of AsA and GSH [20]. The antioxidative role of GSH is closely related to the minimization of ROS, the reduction in dehydroascorbate (DHA), zeaxanthin, and α-tocopherol [20]. As one of the most abundant non-enzymatic antioxidants, GSH plays a central role in maintaining a plant redox state stability and in defense responses to various types of stress [21]. The function of the most common and potent antioxidant, AsA, is to prevent and reduce the damage to plants caused by ROS. Furthermore, it can directly scavenge superoxide anion (O_2_^•−^), singlet oxygen (^1^O_2_), hydroxyl radical (OH^•^), and regenerate oxidized carotenoids (Car) or α-tocopherol [16,22]. AsA also plays an important role in scavenging ROS in chloroplasts [23].

Pathogen attack can alter many primary metabolic processes, such as photosynthesis [24]. During plant–pathogen interactions, the demand for assimilates is increased, and the production of ROS is particularly high and is mainly generated by photosynthetic reactions [24,25]. This explains why some studies have reported on the involvement of photosynthesis in the immune defense of wheat against fungal pathogens [26]. Except for chlorophyll, Car also play an important role in photosynthesis, photo-oxidative protection, and biosynthesis of phytohormones such as abscisic acid (ABA) [27]. To track the effects of pathogen infection on photosynthesis, photosynthetic pigments analysis and chlorophyll *a* (Chl *a*) fluorescence are commonly used. Chl *a* fluorescence is a method that is a sensitive and non-invasive marker of photosynthetic efficiency and as such, is even used for early detection of pathogen infection when symptoms are not yet visible [24].

Phytohormones are known to play a crucial role in defending wheat against various pathogens, including *Fusarium* spp. [28]. Among them, salicylic acid (SA) regulates various parts of wheat growth and development but also plays a key role in wheat resistance to *Fusarium graminearum* [14,29,30]. SA plays a role as a signaling and intermediate molecule in the response to pathogen attack. It is required for systemic acquired resistance (SAR) [14,30,31,32], in which wheat plants produce signals to activate the resistance mechanisms in uninfected parts of wheat plants [33]. SAR also triggers the expression of genes encoding pathogenesis-related (PR) proteins [31,33], which are used as SAR markers in various plants [34]. While the role of SA in activating resistance mechanisms in response to pathogen attack is already well known and described, some studies indicate that ABA is a phytohormone that regulates responses to biotic stresses in addition to its role in developmental processes and acclimation to abiotic stresses [35,36,37]. Although these studies suggest that ABA can positively or negatively affect plant resistance depending on the pathosystem, in most cases, increasing ABA was found to negatively affect plant resistance [38], especially in wheat and barley plants infected with pathogenic fungi [37]. The ABA-modulated resistance to fungal pathogens is thought to depend on several factors, such as an increase in the intensity of this hormone and the type of fungal pathogen [37]. Most evidence suggests that ABA influences pathogen invasion and spread, as well as the expression of certain genes associated with plant responses to biotic stress, and thus acts as a fungal effector [36,37,39]. However, the exact mechanisms behind this role are not yet fully understood.

As wheat plants are repeatedly exposed to various pathogens, they have developed sophisticated mechanisms to control responses to these pathogens. Different previous studies focused on the antioxidant response of winter wheat to FHB stress in field conditions, where, except for FHB, various kinds of abiotic and biotic stresses can interact. Furthermore, winter wheat response to FHB stress was studied in the controlled conditions, too. Significant attempts have already been made to pinpoint mechanisms included in wheat defense response to FHB. However, there is still much to clarify about these mechanisms. Therefore, this study aimed to extend previous studies on the physiological and biochemical responses of winter wheat to FHB stress in a controlled environment by focusing on the AsA-GSH pathway, photosynthetic efficiency, and stress hormone levels, thus providing insight into the possible interactions of different defense mechanisms during infection. Since these cultivars have already been characterized for FHB resistance under field conditions, in this study we hypothesize that varieties differing in resistance to FHB will also show differences in the above-mentioned stress hormones, AsA-GSH metabolism, and key parameters of chlorophyll fluorescence and photosynthetic pigments.

## 2. Results

### 2.1. Disease Severity

In order to assess disease severity, the number of infected spikelets per spike was counted at 10 dpi (days post inoculation), before tissue sampling. Symptoms of the infection were not visible earlier than 7 dpi when the majority of spikes exhibited FHB symptoms. At 10 dpi, almost all varieties had clearly visible spike bleaching (Figure 1). Table 1 presents an average number of infected spikelets for each variety at 10 dpi. As can be seen, variety Golubica had the highest disease severity, with an average number of infected spikelets of 3.7. Tika Taka and El Nino had lower disease severity, compared to Golubica, which was 3.5 and 2.5, respectively. In varieties Kraljica and Galloper, disease spread was even lower compared to the previous varieties, with an average number of infected spikelets per spike of 2 for Kraljica and 1.8 for Galloper. Variety Vulkan had the lowest disease severity, with only 1 spikelet exhibiting FHB symptoms at 10 dpi.

### 2.2. GSH and GSSG Content

GSH content in spikes of tested wheat varieties was significantly affected by FHB stress only in varieties Tika Taka and Golubica (Figure 2a). These varieties significantly increased GSH concentration, compared to the control, by 29.3% and 15.8%, respectively. In the FHB-stressed spikes of the varieties Galloper and El Nino GSH content increased non-significantly compared to controls.

All varieties tested showed an increasing trend of oxidized GSH (GSSG) in FHB treatment, compared to the corresponding controls, where the GSSG concentration increase was not significant only in the variety Vulkan (Figure 2b). The highest significant increase in GSSG content was found in variety Golubica (99.9%), while the lowest significant increase was found in variety Galloper (21.3%).

### 2.3. Activities of the AsA-GSH Cycle Enzymes

APX activity was significantly increased in the FHB-stressed spikes of varieties Galloper, Kraljica, and Vulkan, compared to corresponding controls, with the highest increase recorded in varieties Kraljica (92.4%) and Galloper (73.4%) (Figure 3a). Varieties Tika Taka and Golubica in FHB treatment significantly decreased APX activity, compared to the control, for 78.5% and 51.7%, respectively. A non-significant change was observed in the APX activity of the variety El Nino.

FHB stress significantly decreased MDHAR activity in spikes of four tested wheat varieties, compared to controls (Figure 3b). Variety Golubica had the highest decrease in MDHAR activity (33.5%), followed by varieties Galloper (29%), Tika Taka (26.9%), and El Nino (19.5%). Stress induced by *Fusarium* infection significantly increased MDHAR activity of variety Kraljica (49.6%), compared to the control, while non-significant decreasing trend was detected in variety Vulkan.

DHAR showed a similar trend in its activity as MDHAR. Most of the varieties significantly decreased DHAR activity as a response to FHB stress, compared to the corresponding control (Figure 3c). However, in inoculated spikes of varieties Kraljica and Vulkan, a significant increase in DHAR activity was recorded. The highest decrease in DHAR activity of 38.4% was found in variety El Nino, followed by Golubica (30.6%), Galloper (29.9%), and Tika Taka (21.1%).

*Fusarium* inoculations significantly induced GR activity in spikes of five out of the six varieties tested, compared to controls (Figure 3d). The most prominent increase in GR activity induced by FHB stress was recorded in variety Kraljica (65.2%), while varieties El Nino, Tika Taka, Golubica, and Galloper increased it for 49.3%, 39.1%, 23.6%, and 20.6%, respectively. A non-significant change in GR activity in inoculated spikes of the variety Vulkan was recorded compared to the control.

### 2.4. Pigments and Photosynthetic Efficiency

The Chl *a* did not show a uniform increasing or decreasing trend in its content among varieties (Figure 4a). Varieties Tika Taka and Vulkan significantly decreased Chl *a* in FHB treatment, compared to control, for 12.5% and 17.6%, respectively. However, variety Golubica significantly increased Chl *a* content in FHB-stressed spikes by 14.2%, compared to control. In FHB-stressed spikes of varieties Galloper, Kraljica, and El Nino, Chl *a* content did not change significantly.

Chl *b* content showed the same trend as for Chl *a* (Figure 4b). Varieties Tika Taka, Galloper, and Vulkan decreased Chl *b*, while only Tika Taka and Vulkan decreased it significantly for 27% and 16%, respectively. Varieties Kraljica, Golubica, and El Nino increased Chl *b*, although this increase was significant only in variety Golubica (20.9%).

Car concentration decreased significantly in FHB-stressed spikes of varieties Kraljica (12.9%) and Vulkan (20.2%) compared to control, while the decrease in variety Galloper was not significant. Varieties Golubica and El Nino significantly increased Car in inoculated spikes by 19.3% and 11.9%, respectively. Variety Tika Taka also increased Car but non-significantly (Figure 4c).

Significant changes in chlorophyll *a* to chlorophyll *b* ratio (Chl *a*/Chl *b*) were found in varieties Tika Taka, Kraljica, and Vulkan (Figure 4d). In FHB-stressed spikes of previously mentioned varieties, Chl *a*/Chl *b* significantly increased in variety Tika Taka (22.7%) and decreased in varieties Kraljica (16%) and Vulkan (7%) compared to control ones.

Carotenoids-to-total-chlorophyll ratio (Car/Chl *a* + Chl *b*) significantly decreased only in varieties Galloper (4.6%) and Kraljica (13.6%), while in the rest of the varieties this ratio increased (Figure 4e). The increase in Car/Chl *a* + Chl *b*, which was the highest and at the same time only significant, was recorded in FHB-stressed spikes of variety Tika Taka (40.1%). The lowest increase in Car/Chl *a* + Chl *b* ratio was obtained in the case of variety Vulkan (2.3%).

Changes in maximum quantum yield of primary photochemistry (TR_0_/ABS) in control and FHB-stressed spikes on four measurement points (1 dpi, 3 dpi, 7 dpi, and 10 dpi) of each variety separately are shown in Figure 5. On the first two measurement points (1 dpi and 3 dpi), FHB inoculations, compared to controls, did not significantly affect TR_0_/ABS in six investigated varieties (Figure 5a–f). However, in the variety Golubica, a significant sharp decline of TR_0_/ABS in FHB treatment was recorded at 7 dpi and 10 dpi, compared to 3 dpi (93.4% and 96.5%) (Figure 5a). When comparing differences between FHB treatment and controls, a significant decrease in TR_0_/ABS in FHB-stressed spikes was observed at 7 dpi and 10 dpi (93.7% and 96.4%). This was the same as for the Golubica variety, where a significant decrease in TR_0_/ABS was recorded in the inoculated spikes of variety Tika Taka at 7 dpi (96.3%) and 10 dpi (92.5%), compared to controls (Figure 5b). Furthermore, in the FHB-inoculated treatment, TR_0_/ABS measured at 7 dpi and 10 dpi, compared to 3 dpi, and was significantly decreased by 96.3% and 92.7%, respectively. Variety El Nino had approximately the same response as for varieties Tika Taka and Golubica. Drastic changes in FHB inoculated treatment were observed in spikes at 7 dpi and 10 dpi, where TR_0_/ABS significantly decreased when compared to 3 dpi (69.5% and 93.9%) (Figure 5c). Furthermore, significant declines were recorded in FHB-stressed spikes at 7 dpi (69.2%) and 10 dpi (93.4%) when compared to controls. In the FHB inoculated treatment, a significant decrease in TR_0_/ABS in spikes of the variety Kraljica at 7 dpi (61.3%) and 10 dpi (49.3%) was recorded, compared to 3 dpi, too (Figure 5d). However, a significant change between control and FHB-stressed spikes was obtained only at the third measurement point (7 dpi), where TR_0_/ABS decreased by 63.1%. In Galloper, a significant decrease in TR_0_/ABS was recorded as a consequence of FHB inoculations at 10 dpi by 61.8%, compared to controls (Figure 5e). When observing measurement points in FHB treatment, it is obvious that TR_0_/ABS was significantly decreased in the last two measurements, compared to the second measurement by 38.4 and 64.2%, respectively. Observed changes in TR_0_/ABS in spikes of the Vulkan variety were much less prominent, compared to the rest of the varieties studied (Figure 5f). When comparing control and FHB-stressed spikes at each measurement point separately, a significant decline was found only at 7 dpi (24.9%) in FHB-treated spikes, compared to control. A significant decrease in this parameter in FHB treatment was observed at the third (7 dpi) and fourth (10 dpi) measurement points when compared to the second measurement (3 dpi) by 23.9% and 28.3%, respectively.

Changes in performance index on absorption basis (PI_abs_) in control and FHB-stressed spikes of each variety separately are presented in Figure 6. There were no significant changes in FHB inoculated treatment at the first two measurement points (1 and 3 dpi), while in the last two measurements (7 and 10 dpi), a significant decline of PI_abs_ can be observed in almost all varieties studied, compared to the first two measurements (Figure 6a–f). The exception is, however, Galloper variety, where in FHB treatment, a significant decrease was observed at 7 dpi, compared to 1 dpi, and at 10 dpi, compared to 1 and 3 dpi (Figure 6d).

When comparing PI_abs_ in control and FHB-stressed spikes of each measurement point separately, a significant decrease of 100% and 97.6% in FHB-stressed spikes was recorded in variety Tika Taka (Figure 6b), 99.6% and 100% in variety Golubica (Figure 6a), and 80.1% and 99.5% in variety El Nino (Figure 6c) at 7 dpi and 10 dpi, respectively. Varieties Kraljica and Vulkan significantly decreased this parameter in FHB-inoculated spikes only at 7 dpi for 90.7% and 57.6%, respectively, compared to controls (Figure 6d,f). FHB inoculations did not cause significant changes in FHB-stressed spikes of these varieties at 10 dpi, compared to controls. FHB inoculations in variety Galloper, compared to controls, significantly affected PI_abs_ only at the last measurement point (10 dpi) by 75.4% (Figure 6e).

### 2.5. ABA and SA Content

*Fusarium* infection significantly increased ABA content in all spikes of winter wheat varieties tested, compared to corresponding controls. The highest increase was observed for varieties Golubica (485.3%) and Tika Taka (453.9%), while the lowest increase was recorded for variety Kraljica (48.4%) (Figure 7a).

The content of SA in spikes of tested varieties was differentially affected by artificial *Fusarium* infections, compared to controls. Significant changes in SA content were observed only for variety El Nino, which showed a significant increase in SA content (54.6%), and variety Vulkan, which showed a significant decrease in SA content (34.2%) in FHB treatment, compared to controls (Figure 7b).

### 2.6. Principal Component Analysis (PCA) Analysis between Investigated Traits

The PCA biplot showed that 39.1% of the total variability was explained by the first principal component (PC1) and 24.0% by the second principal component (PC2) (Figure 8). The first two principal components (PCs) together explained 63.1% of the total variability. As can be seen from the biplot, five wheat varieties from FHB treatment were grouped on the left side (except Vulkan that was outside of FHB cluster), while all wheat varieties in controls were on the right side of the PCA plot, indicating a negative correlation between varieties from FHB treatments and controls. At the same time, GSSG, GSH, GR, ABA, and type II resistance were grouped on the left side of the PCA plot, while MDHAR, DHAR, TR_0_/ABS, PI_abs_, and APX were shifted toward the right side of the PCA plot. Positively related were Chl *a*, Chl *b* and SA that further were in negative relation with Car/Chl *a* + Chl *b* and Chl *a*/Chl *b*.

## 3. Discussion

The world population and accompanying food needs are constantly increasing, while wheat and other cereals contribute significantly to the global food supply [40]. It is already known that FHB is a serious wheat fungal disease that results in massive grain yield and quality losses and represents a threat to food security. However, knowledge of the mechanisms that participate in defense reactions is scarce, and growing resistant varieties is considered the most efficient tool for managing FHB [4]. To better understand winter wheat resistance mechanisms to FHB, we investigated changes in the crucial part of the antioxidant system, the AsA-GSH cycle. Furthermore, we extended our investigations to the level of endogenous stress hormones ABA and SA, and to the photosynthetic efficiency of varieties differing in FHB resistance. Our main focus was on changes in wheat spikes as the primary site of FHB infection.

The early stages of *Fusarium* infection up to the fifth day after FHB inoculation were reported to be asymptomatic. After a few days, as mycelia spread up and down the rachis and through the spikelets, the spike axis was degraded, and symptoms became visible as bleaching [28,41]. Studies report different types of active FHB resistance mechanisms [28]. These include resistance to initial infection (type I), resistance to spread of the pathogen in infected tissues (type II), resistance to mycotoxins (type III), resistance to kernel infection (type IV), and grain yield tolerance (type V). The first two types of resistance to FHB as well as resistance to mycotoxins are widely accepted, but the only type that is frequently characterized is type II resistance [42]. Since type II resistance is determined by a central spikelet inoculation in greenhouse experiments [43], the same has been performed in the present study. According to the results, varieties Golubica and Tika Taka were characterized as FHB susceptible, while variety El Nino possessed low type II resistance and could be characterized as FHB moderately susceptible. According to the number of infected spikelets at 10 dpi, Kraljica and Galloper were characterized as moderately resistant varieties, while variety Vulkan possessed high type II resistance and was identified as FHB resistant.

### 3.1. AsA-GSH Metabolism Response to FHB Stress

Following fungus detection by different proteins and cell wall components, several defense mechanisms are induced, including activation of non-enzymatic and enzymatic antioxidants [16,44]. The determination of the oxidative status and antioxidative response of wheat spikes following *Fusarium* inoculations in the present study was conducted by measuring the content of the GSH and GSSG, as well as the activities of the enzymes of the AsA-GSH pathway (APX, MDHAR, DHAR, and GR). GSH, together with AsA, is one of the crucial factors in stress tolerance of different plants, including wheat [20]. Higher GSH content in the present study mostly had varieties characterized as FHB susceptible or moderately susceptible. The GSH content of the varieties Kraljica and Vulkan was only slightly changed compared to the other varieties in this study. In contrast, GSSG content was significantly higher in all varieties except Vulkan. Significantly higher GSSG concentrations in almost all varieties in this study suggest lower and insufficient GSH recycling under the stress induced by *Fusarium* inoculations, despite increased GR activity. Increased GSH contents in FHB susceptible varieties could also be due to de novo GSH synthesis, indicating increased oxidation of cytosol [45].

The pools of GSH and AsA are strongly correlated with the activities of the enzymes of the AsA-GSH pathway. APX is a potent scavenger of H_2_O_2_ [20]. In the present study, APX activity increased in Galloper, Kraljica, and Vulkan, with the most pronounced increase in Kraljica and Galloper. Severe stress caused by FHB inoculations in susceptible varieties Tika Taka and Golubica significantly reduced APX activity. Similar results were obtained in the field conditions, where APX was reduced immediately after inoculations in the FHB inoculated susceptible variety, compared to the non-inoculated [12]. During APX detoxification of H_2_O_2_, AsA is oxidized to monodehydroascorbate (MDHA). MDHA is then enzymatically recycled back to AsA by the activity of the enzyme MDHAR [46]. In our study, MDHAR activity was decreased in the inoculated spikes of all varieties except Kraljica, with the most prominent decrease in the variety Golubica. MDHA can also non-enzymatically disproportionate to AsA and dehydroascorbate (DHA) [46]. Therefore, these results indicate that MDHA is not successfully recycled to AsA by MDHAR, and consequently, DHA is formed. The next important role in the AsA-GSH pathway is played by another ascorbate reductase, DHAR, which is responsible for the reduction in DHA back to AsA using GSH as an electron donor [47]. DHAR activity showed a similar trend as MDHAR in the present study. The activity of this enzyme in *Fusarium* inoculated spikes was mostly decreased. However, FHB moderately resistant Kraljica and resistant Vulkan increased its activity, which implies successful AsA recycling. Except for AsA, GSSG produced during the detoxification of ROS by APX needs to be recycled back to GSH. GSH is regenerated by the flavoprotein oxidoreductase GR, mainly localized in chloroplasts [22]. Contrary to other enzymes in our study, stress caused by FHB inoculations increased GR activity in spikes of all varieties, implying higher GSH depletion and the need for GSH recycling, except for the FHB-resistant variety Vulkan. Khaledi et al. [16] measured the activities of the antioxidant enzymes in leaves and spikes inoculated with *F. graminearum* and *F. culmorum* in two varieties differing in resistance level to FHB and in corresponding control tissue. Results showed that activities of all measured enzymes increased from the flowering stage until the milk stage in moderately resistant, compared to susceptible variety, in both treatments. Nevertheless, increases in the activity of APX are in accordance with the results obtained in our study, where APX activity increased in FHB-inoculated spikes of all moderately resistant and resistant varieties. Similar results were obtained in the study by Motallebi et al. [48], where the authors reported increased enzyme activities in the seedlings of resistant wheat varieties infected with *F. culmorum*. Furthermore, the authors observed much earlier induction of the antioxidant enzymes in moderately resistant and resistant varieties, followed by a decrease at 6 dpi. Similar results were obtained in other studies too [49,50,51]. These early inductions of different antioxidant enzymes in wheat varieties possessing FHB resistance could potentially explain why enzyme activities in FHB-inoculated spikes of moderately resistant and resistant varieties in our study were decreased at 10 dpi when disease symptoms already showed up. Overall, the results obtained in this study suggest that the AsA-GSH pathway plays an important role in the response to FHB.

### 3.2. Pigments and Photosynthetic Efficiency of Wheat Spikes in Response to FHB Stress

In addition to enzymatic and non-enzymatic systems, FHB inoculations can also alter photosynthetic pigments. It is a generally accepted fact that high oxidative damage reduces the synthesis and accumulation of chlorophylls and other photosynthetic pigments [52]. In our study, elevated concentrations of Chl *a* were recorded in FHB-inoculated spikes of varieties Golubica and El Nino, while in the rest of the varieties, its concentration was decreased or non-significantly affected by stress induced by FHB inoculations. Similar results were obtained for Chl *b*, too. Chl *b* contents were increased in Golubica and El Nino, but also in the variety Kraljica. Recent studies suggest that chlorophylls respond differently to environmental stresses and are an important part of the signal transduction network associated with stress responses. More precisely, low-dose stress can inhibit chlorophyll degradation and stimulate chlorophyll biosynthesis [53,54], thereby mitigating stress in several ways including the production of carbon-based defense molecules [55]. Car have been reported to be part of the non-enzymatic antioxidant system due to their redox potential and as such protect cells from different environmental stresses [56,57]. The same trend of pigment increase in *Fusarium* inoculated spikes of FHB susceptible and moderately susceptible varieties Golubica, Tika Taka, and El Nino was observed in Car. In addition to an increase in chlorophyll, an increase in Car was also detected in response to low-dose stress [52]. However, our results contradict those obtained by other authors [16] who reported that Car content was higher in leaves and spikes of varieties more resistant to FHB. According to Atanasova-Penichon et al. [58], Car play a crucial role in FHB resistance. Nevertheless, elevated Car content in FHB susceptible varieties may imply the utilization of additional defense strategies in fighting the pathogen. The Chl *a*/Chl *b* ratio showed different trends in our study when compared to Chl *a* and Chl *b* contents. However, this ratio was most pronounced in the FHB-resistant variety Vulkan.

Chlorophyll fluorescence and photosynthetic parameters have already been used for the detection of fungal infections [59,60]. In addition, recent studies suggest that photosynthetic parameters are related to FHB resistance [60]. As indicated by TR_0_/ABS and PI_abs_, severe FHB stress negatively affected photosynthetic efficiency in wheat spikes of almost all varieties tested, but especially of varieties susceptible to FHB. Varieties Tika Taka, Golubica, and El Nino had the highest decrease in TR_0_/ABS and PI_abs_ at 7 and 10 dpi, when compared to corresponding controls. Similar results for tested varieties were obtained in the previous research [61], where the same susceptible varieties showed a decrease in selected JIP-test parameters at the onset of symptom development. Furthermore, the above-mentioned varieties had the highest increase in ABA levels. Results obtained in our studies may imply that ABA is a negative factor of photosynthetic capacity and overall FHB resistance in which Car may have an important role. This assumption is in accordance with other studies, which indicated that the link between inhibition of photosynthesis and environmental stress may be ABA [62]. A possible explanation for this claim could be the fact that ABA promotes stomatal closure, reduces CO_2_ assimilation, and consequently, indirectly inhibits the activity of ribulose 1,5-bisphosphate carboxylase through altered ion fluxes [62].

### 3.3. ABA and SA Response to FHB Stress

Considering endogenous stress hormones, ABA is primarily important for eliciting adaptive responses to different types of abiotic stresses, such as drought, low temperature, and salinity [63]. However, there is increasing evidence that ABA interferes with other shared components of biotic stress signaling [64]. The concentration of ABA increased in all varieties studied, with the highest increase recorded in FHB susceptible varieties Golubica and Tika Taka. These results are consistent with some previous studies in which inoculation with *F. graminearum* increased levels of endogenous hormones such as ABA and related metabolites, SA, jasmonic acid (JA), and auxins (IAA) 4 days after inoculation [36]. Although research on ABA concentration following inoculations of wheat with *Fusarium* species is rare, Qui et al. [65] investigated transcriptomic changes of several phytohormones in wheat spikes, including ABA. The research, which resulted in reference maps of hormone responses, facilitated the investigation of the roles of the hormone pathways in wheat responses to biotic stress, particularly FHB. The authors concluded that inducing the accumulation of ABA after a pathogen attack most likely suppresses FHB tolerance by inhibiting the expression of phenylalanine pathway genes. Consequently, suppression of phenylalanine pathway genes leads to inhibition of flavonoid and lignin biosynthesis and weakening of physical barriers to the fungus [65]. These results are consistent with those of our study, in which two winter wheat varieties exhibiting the highest occurrence of FHB symptoms, Golubica and Tika Taka, had the most prominent increase in ABA level following *Fusarium* inoculations.

The SA-mediated signaling pathway is one of the crucial pathways in wheat resistance to FHB [14]. Contrary to ABA, SA showed an inconsistent response to *Fusarium* inoculations. Almost all of the varieties studied increased SA concentration as a response to FHB infection, except varieties Tika Taka and Vulkan, which decreased it. It is still unclear how SA affects wheat defense mechanisms to control FHB [31]. Some studies have reported that many *Fusarium* species have the ability to metabolize SA through the enzyme salicylate hydroxylase [66,67]. This could probably be the reason why exogenous SA treatments of wheat spikes do not affect FHB resistance [31,68]. However, there is evidence that the relationship between SA and ROS, particularly H_2_O_2_, is complicated [69]. Rao et al. [70] showed that treatments of *Arabidopsis thaliana* with exogenous SA resulted in increased H_2_O_2_ content. Some authors pointed out that the induced production of SA is particularly important in the very early stages of FHB infection of wheat [29,71]. Ding et al. [29] reported a biphasic phenomenon in the first 24 h after wheat inoculation with *F. graminearum*, where the SA and Ca^2+^ pathways were activated within 6 hours after inoculations, followed by the activation of the JA-mediated pathway about 12 h after inoculation. Similar findings were confirmed at the gene level by Ameye et al. [71] who observed a strong up-regulation of genes for SA biosynthesis at 24 h after inoculation, followed by its down-regulation in the next 24 h. Furthermore, Spanic et al. [51] concluded that a rapid induction of H_2_O_2_ concentration in spikes of resistant winter wheat varieties in the early stage of FHB infection could be the reason of improved FHB resistance, considering the fact that H_2_O_2_ acts as a signal molecule for induction of SAR. These results could potentially explain why the SA content in the present study was lower or just slightly changed at 10 dpi.

### 3.4. Relations between Analyzed Traits in FHB Treatment of Six Winter Wheat Varieties and Their Respective Controls

Correlation of different traits and varieties under FHB stress conditions presented by PCA clearly showed that Vulkan is the most FHB resistant variety. Further, GSSG, GSH, GR, and ABA were closely located to FHB susceptible and moderately susceptible varieties in FHB treatment. Those traits and varieties were in negative relation with TR_0_/ABS, PI_abs_ and APX mostly showing the reduction in FHB resistant and moderately resistant varieties in FHB treatment, compared to controls. Those relations confirmed the previous statement about AsA-GSH pathway as one of the major components of the antioxidant response to FHB stress influencing functionality of photosystem II. The close distance between ABA and Car showed the importance of their mutual relation and involvement in FHB susceptibility, especially in FHB-inoculated spikes of FHB susceptible varieties. Still, it is not clear how SA influenced Chl *a* and Chl *b*, as we might assume that response of SA to FHB in wheat spikes might have occurred before 10 dpi.

## 4. Materials and Methods

### 4.1. Plant Material and Growth Conditions

The total of six winter wheat varieties used in this research (Golubica, Tika Taka, El Nino, Galloper, Kraljica, and Vulkan) originated from the Agricultural Institute Osijek. Seeds of each winter wheat variety were first sown in seedlings’ trays and placed at room temperature to germinate. Trays with five-day-old winter wheat seedlings were moved in a plant growth chamber for six weeks to undergo a period of vernalization (12/12 h 40% light intensity/dark photoperiod, 4/3 °C day/night temperature, and 60% relative humidity). Plants were then transferred in 2.5 L pots filled with soil (pH: 5.5–7.0, organic matter: 70.0–85.0%, N (1/2 vol.): 100–200 mg L^−1^, P_2_O_5_ (1/2 vol.): 100–150 mg L^−1^, K_2_O (1/2 vol.): 200–400 mg L^−1^) and placed in a greenhouse (Gis Impro d.o.o., Vrbovec, Croatia). During tillering stage (GS21), the conditions maintained in the greenhouse were 10/14 h light/dark photoperiod, 10–14/8–12 °C day/night temperature with the maximum light intensity of 250 μmol m^−2^ s^−1^. With the start of stem elongation stage (GS31), conditions were set up to be 12/12 h light/dark photoperiod and 15–18/11–14 °C day/night temperature, while before anthesis stage (GS51) and until the end of experiment conditions were set up to 14/10 h light/dark photoperiod, 21–24/17–20 °C day/night temperature with the maximum light intensity of 750 μmol m^−2^ s^−1^. During the whole experiment plants were irrigated with water as necessary, usually twice per week. Nitrogen fertilization was carried out at the two-leaf development stage (GS12) using calcium ammonium nitrate (CAN) (27% N) and one fertilization against pests with the insecticide Vantex (gamma-cyhalothrin 60 g L^−1^) (GS31). When the anthers started to extrude and flowering stage appeared (GS61), plants were inoculated with a mixture of *Fusarium graminearum* and *F. culmorum* inoculum. Each treatment consisted of six replicates set up in a randomized complete block design, where each replicate contained 4 plants/pot. Untreated plants were used as controls. Ten days after inoculation spike tissue for determination of GSH and GSSG, activities of the enzymes of the AsA-GSH cycle, and photosynthetic pigments was sampled, frozen in liquid nitrogen, and stored at −80 °C prior to further analysis. Before extractions, wheat spike tissue was grounded in a 10 mL stainless steel jars containing a grinding ball for 1 min at 30 Hz using a TissueLyser (Qiagen Retsch GmbH, Hannover, Germany). For the analysis of stress hormones, ABA and SA, sampled wheat spikes were frozen in liquid nitrogen and lyophilized.

### 4.2. Inoculation of Wheat Spikes and Disease Severity Assessment

Two *Fusarium* species were used for wheat inoculation: *F. graminearum* (PIO 31), isolated from the winter wheat collected in the eastern part of Croatia, and *F. culmorum* (IFA 104) obtained from IFA-Tulln, Austria. The conidial inoculum of *F*. *graminearum* and *F. culmorum* was produced by a mixture of wheat and oat grains (3:1 *v/v*). Macroconidia were washed off the colonized grains, suspension was diluted, final conidial concentrations of both fungi were set to 5 × 10^4^ mL^−1^ using a hemocytometer (Bürker-Türk, Hecht Assistent, Sondheim vor der Rhön, Germany) and diluted suspension of *F. graminearum* and *F. culmorum* were mixed together. The 20 μL of prepared inoculum mixture was injected with an automatic pipette (Eppendorf, Wien, Austria) in the middle spikelet of the spike of each plant. Plants were subjected to two inoculation events two days apart. To provoke infection, misting treatment started one hour after each inoculation and lasted for the next 36 hours, where foggers sprayed the water every hour for a period of two minutes.

Type II resistance to FHB (resistance to disease spread within the spike) was evaluated by counting the number of infected spikelets in the inoculated spike of one plant/pot at 10 dpi.

### 4.3. Determination of the Glutathione Content

Total (tGSH) and oxidized glutathione (GSSG) content were determined using a kinetic method based on a continuous reduction in 5,5-dithiobis (2-nitrobenzoic acid) (DTNB) to 5-thio-2-nitrobenzoic acid (TNB) by reduced glutathione (GSH), where GR and NADPH reduce the GSSG [72], modified for the microplate assay. For tGSH content determination, the frozen wheat spike tissue powder was homogenized with 5% 5-sulfosalycilic acid solution (1/10 *w/v*) and centrifuged for 15 min at 16,000× *g* and 4 °C. The reaction mixture consisted of 10 μL of resulting supernatant, 0.03 mg mL^−1^ DTNB, 0.11 U mL^−1^ GR, 1 mM EDTA, and 100 mM phosphate buffer (pH 7.0) in a final volume of 0.21 mL. Following a 5-min equilibration period, NADPH in a final concentration of 0.04 mg mL^−1^ was added to initiate a reaction. The formation of TNB was continuously recorded at 412 nm for 5 min at 25 °C. The amount of tGSH was determined by a standard curve of GSH, and results were expressed as nmol g^−1^ of FW. For GSSG determination, 2% of vinylpyridine and 5% of triethanolamine were added to an aliquot of deproteinized supernatant, and the reaction mixture was incubated for one hour at room temperature. The measurements were performed the same way as for the tGSH. The content of GSSG was determined using a standard curve for GSSG, and the results were expressed as nmol g^−1^ of FW. From the difference between tGSH and GSSG, the GSH content was obtained and expressed as nmol g^−1^ of FW.

### 4.4. Activities of the Enzymes of Ascorbate–Glutathione Cycle

The wheat spike tissue powder obtained by grounding was homogenized with a cold 100 mM phosphate buffer (pH 7.0) containing 1 mM EDTA. Homogenized samples were then incubated for 15 min on ice and centrifuged for 15 min at 19,000× *g* and 4 °C. Aliquots of obtained protein extracts were stored at −80 °C until further analysis. Additionally, protein concentration in the enzyme extracts was determined using bovine serum albumin as a protein standard [73]. Measurements were performed using 96-well plates on a Spark Multimode microplate reader with SparkControl software version 2.1 (Tecan, Männedorf, Switzerland).

APX (EC 1.11.1.11) activity was measured according to a method described by Nakano and Asada [74]. The reaction mixture (0.205 mL) consisted of 0.6 mM AsA, 5 mM H_2_O_2_, and diluted protein extract in 50 mM potassium phosphate buffer (pH 7.0). After 3 min of incubation at room temperature, the decrease in absorbance was measured at 290 nm for 5 min every 15 s. The APX activity was calculated using the molar extinction coefficient (ε = 1.708 mM cm^−1^) and expressed in U g^−1^ of protein.

MDHAR (EC 1.6.5.4) was determined according to a method described by Hossain et al. [75] and adjusted for a microplate assay. The reaction mixture consisted of 50 mM Tris-HCl buffer (pH 7.8), 0.45 mM NADH, 2.25 mM AsA, and diluted protein extract in a final volume of 0.2 mL. After equilibration at room temperature, the reaction was started by adding ascorbate oxidase in a final concentration of 0.14 U mL^−1^. The decrease in absorbance was monitored at 340 nm for 3 min. MDHAR activity was calculated using the molar extinction coefficient (ε = 3.7 mM cm^−1^) and expressed as U g^−1^ of protein.

DHAR (EC 1.8.5.1) activity was determined by a method based on monitoring the GSH-dependent reduction in DHA described by Ma and Cheng [76] and adjusted for a microplate assay according to Murshed et al. [77]. The reaction mixture consisted of 50 mM HEPES buffer (pH 7.0), 0.09 mM EDTA, 2.25 mM GSH and 0.2 mM DHA (0.2 mL). The increase in absorbance was recorded at 265 nm for 3 min. DHAR activity was calculated using the molar extinction coefficient (ε = 8.33 mM cm^−1^) and expressed as U g^−1^ of proteins.

GR (EC 1.6.4.2) activity was measured according to a method by Racker et al. [78] and adjusted for microplate assay by Murshed et al. [77]. The reaction mixture consisted of 50 mM HEPES buffer (pH 8.0), 0.45 mM EDTA, 0.23 mM NADPH, and protein extract in a final volume of 0.2 mL. After 10 min of equilibration at room temperature, the reaction was started by adding GSSG in a final concentration of 0.5 mM. The decrease in absorbance was monitored at 340 nm for 5 min every 15 s. GR activity was calculated using the molar extinction coefficient for NADPH (ε = 3.7 mM cm^−1^) and expressed in U g^−1^ of protein.

### 4.5. Photosynthetic Pigments

Photosynthetic pigments were measured according to a method by Lichtenthaler [79]. Fresh wheat spike tissue powder was homogenized in absolute acetone, followed by extraction for 15 min at 4 °C and centrifugation for 15 min at 16,000× *g* and 4 °C. The procedure was repeated until the plant material was uncolored. The chlorophyll and Car content was determined spectrophotometrically, and photosynthetic pigment concentrations were expressed as mg g^−1^ FW.

### 4.6. Measurement and Analysis of Fast Chlorophyll a Fluorescence

The Chl *a* fluorescence (ChlF) measurements on spikes of inoculated and control plants were performed at four measurement points: 1 dpi, 3 dpi, 7 dpi, and 10 dpi. The OJIP fluorescence transients were measured with a Handy-PEA fluorimeter (Plant Efficiency Analyzer, Hansatech Instruments Ltd., King’s Lynn, Norfolk, UK). For each of six varieties, six plants from controlled treatment and six plants from FHB inoculated treatment were analyzed by performing measurements on spikes. Before measurements, spikes were dark adapted for 30 min. ChlF transients were induced by pulse of red saturating light (3200 μmol photons m^−2^ s^−1^, peak at 650 nm) and recorded for 1 s. The JIP-test was applied to analyze and compare ChlF transients [80,81].

### 4.7. Abscisic Acid and Salicylic Acid Analysis

Determination of stress hormones (ABA and SA) was performed by liquid chromatography tandem mass spectrometry (LC-MS/MS) as described earlier [63]. In brief, lyophilized wheat spike tissue was grounded using a TissueLyser (Qiagen Retsch GmbH, Hannover, Germany) for 1 min and a frequency of 30 Hz. A powdered sample (30 mg) was extracted in 1 mL of extraction solution (10% MeOH and 1% acetic acid) containing mixture of internal isotope labeled standards SA-d_6_ (Sigma-Aldrich) and (+)-cis, trans ABA-d_6_ (Trc) (final concentration 38.5 ng mL^−1^). After vortexing, the samples were placed in a Mixer Mill (Roche) for 2 min at a 30 Hz frequency, after which they were homogenized for 1 h at 4 °C. The samples were then centrifuged (10 min, 13,000 rpm) and the resulting supernatant was used for analysis. LC-MS/MS screening of target stress hormones was carried out using Agilent Technologies 1200 series HPLC system equipped with 6420 triple quadropole mass spectrometer with electrospray ionization source (ESI) (Agilent Technologies Inc., Palo Alto, CA, USA). Chromatographic separation was performed on the Zorbax XDP C18 column (75 × 4.6 mm, 3.5 μm particle size) (Agilent Technologies Inc., Palo Alto, CA, USA). The electrospray ionization source was operated in negative mode, and samples were detected in the multiple reaction monitoring (MRM) modes. All data acquisition and processing was performed using Agilent MassHunter software (Agilent Technologies, Santa Clara, CA, USA). ABA and SA concentrations were calculated and expressed as ng mg^−1^ of DW.

### 4.8. Data Analysis

The data for determination of GSH and GSSG content, activities of the enzymes of the AsA-GSH cycle, photosynthetic pigments, and parameters of photosynthetic efficiency were presented as mean of six (or four for ABA and SA analysis) independent biological replicates ± standard deviations. Determination of differences among treatments within each variety separately or among treatments of the same measurement point within each variety separately (for TR_0_/ABS and PI_abs_) was performed using two sample *t*-test (*p* < 0.05). Determination of differences among measurement points for TR_0_/ABS and PI_abs_ within each treatment separately was done using one-way analysis of variance (ANOVA), followed by Fisher LSD post hoc test (*p* < 0.05). Data analysis was performed within the R environment version 4.1.2 (R Foundation for Statistical Computing: Vienna, Austria) [82] and Statistica version 12.0 (Statsoft Inc., Tulsa, OK, USA). PCA were plotted by https://www.bioinformatics.com.cn/en (accessed on 21 October 2023), a free online platform for data analysis and visualization.

## 5. Conclusions

FHB resistance is a complex trait comprised of a complex network of defense responses and mechanisms. Although it is known that SA plays a crucial role in FHB resistance, our studies showed conflicting results. The inconsistent levels of SA at 10 dpi in our study may imply that SA as a main factor in wheat resistance to FHB is probably more important in the very early stages of FHB infection. In the present study, increased activities of the enzymes of AsA-GSH pathway were detected in the FHB-inoculated spikes of moderately resistant and resistant winter wheat varieties. This suggests that the AsA-GSH pathway is one of the major components of the antioxidant response to FHB stress, maintaining the redox state of spikes and preserving functional photosystem II, which could be seen as a lower reduction in TR_0_/ABS and PI_abs_ in inoculated spikes of FHB resistant or moderately resistant varieties, compared to controls. Furthermore, Car increased in FHB-stressed spikes of susceptible varieties and decreased in FHB-stressed spikes of resistant and moderately resistant varieties. This may indicate their role as the main pigments in photosystem assembly. Car were also found to play a role in the biosynthesis of ABA, which was significantly increased in FHB-inoculated spikes of susceptible varieties. Although the exact role of ABA in biotic stress response is still ambiguous, increasing evidence suggests its role as a fungal effector whose elevated levels lead to increased FHB susceptibility in inoculated spikes of winter wheat varieties. However, to obtain a clearer picture, future research should be directed on the earlier response of winter wheat to FHB inoculation in a time-course manner at several measurement points as well as on effect of *Fusarium* mycotoxins on the studied parameters.

## Figures and Tables

**Figure 1 plants-12-03720-f001:**
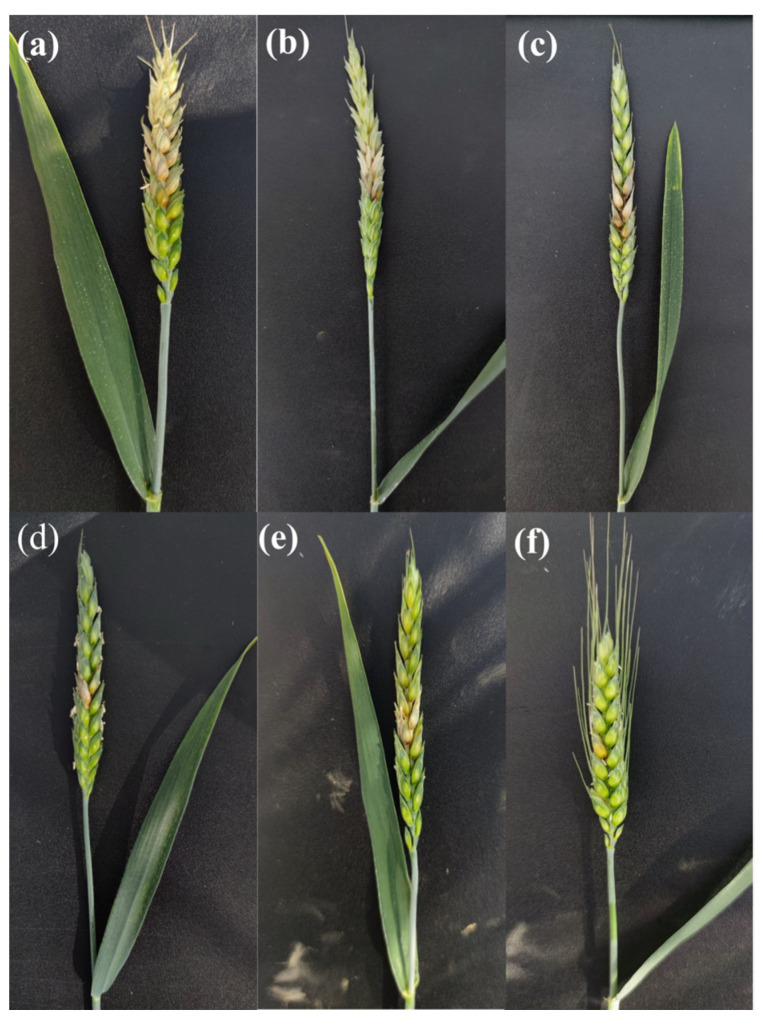
Fusarium head blight (FHB) symptoms of the spike bleaching in variety Golubica (**a**), Tika Taka (**b**), El Nino (**c**), Kraljica (**d**), Galloper (**e**), and Vulkan (**f**) at 10 days post inoculations (dpi).

**Figure 2 plants-12-03720-f002:**
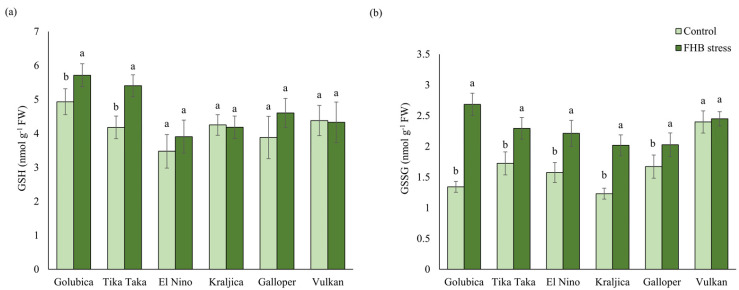
Content of (**a**) reduced glutathione (GSH) and (**b**) oxidized glutathione (GSSG) in the control and FHB-stressed spikes of six winter wheat varieties (Golubica, Tika Taka, El Nino, Kraljica, Galloper, and Vulkan). Bars represent mean values of six independent biological replicates ± SD. Different letters indicate significant difference among treatments in each variety separately (*p* < 0.05).

**Figure 3 plants-12-03720-f003:**
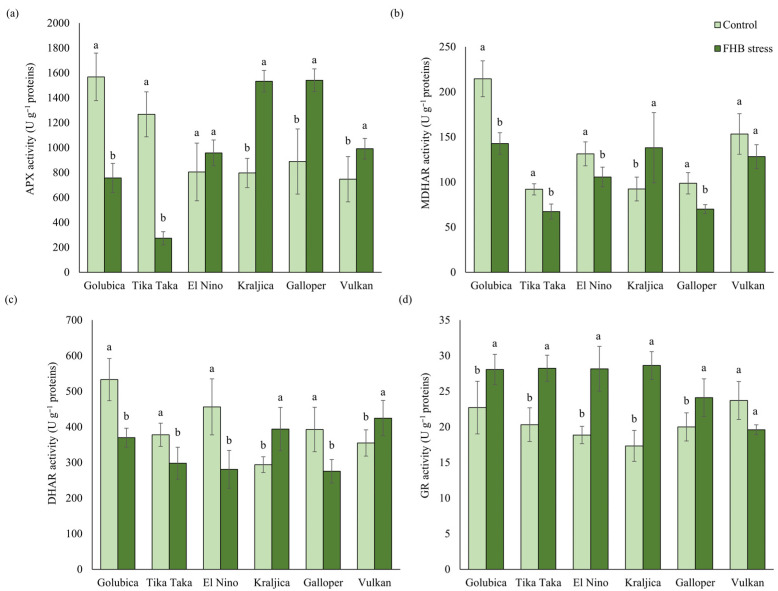
The activity of (**a**) ascorbate peroxidase (APX), (**b**) monodehydroascorbate reductase (MDHAR), (**c**) dehydroascorbate reductase (DHAR), and (**d**) glutathione reductase (GR) in control and FHB-stressed spikes of six winter wheat varieties (Golubica, Tika Taka, El Nino, Kraljica, Galloper, and Vulkan). Bars represent mean values of six independent biological replicates ± SD. Different letters indicate significant difference among treatments in each variety separately (*p* < 0.05).

**Figure 4 plants-12-03720-f004:**
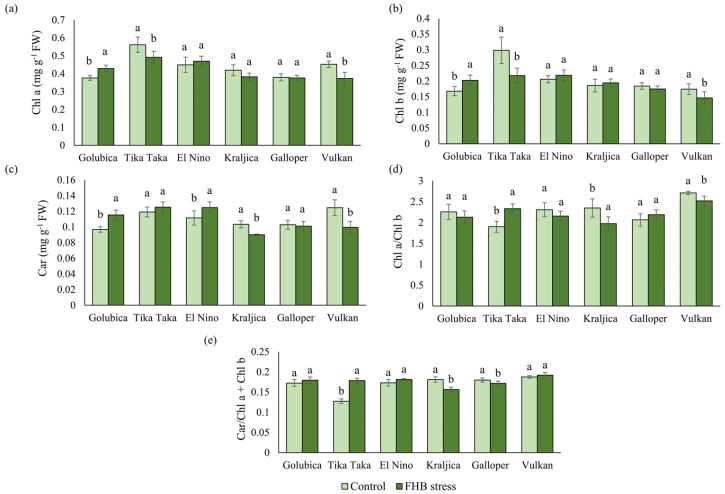
Content of chlorophyll a (Chl *a*) (**a**), chlorophyll b (Chl *b*) (**b**), carotenoids (Car) (**c**), chlorophyll *a*/*b* ratio (Chl *a*/Chl *b*) (**d**), and carotenoids/total chlorophyll ratio (Car/Chl *a* + Chl *b*) (**e**) in control and FHB-stressed spikes of six winter wheat varieties (Golubica, Tika Taka, El Nino, Kraljica, Galloper, and Vulkan). Bars represent mean values of six independent biological replicates ± SD. Different letters indicate significant difference among treatments in each variety separately (*p* < 0.05).

**Figure 5 plants-12-03720-f005:**
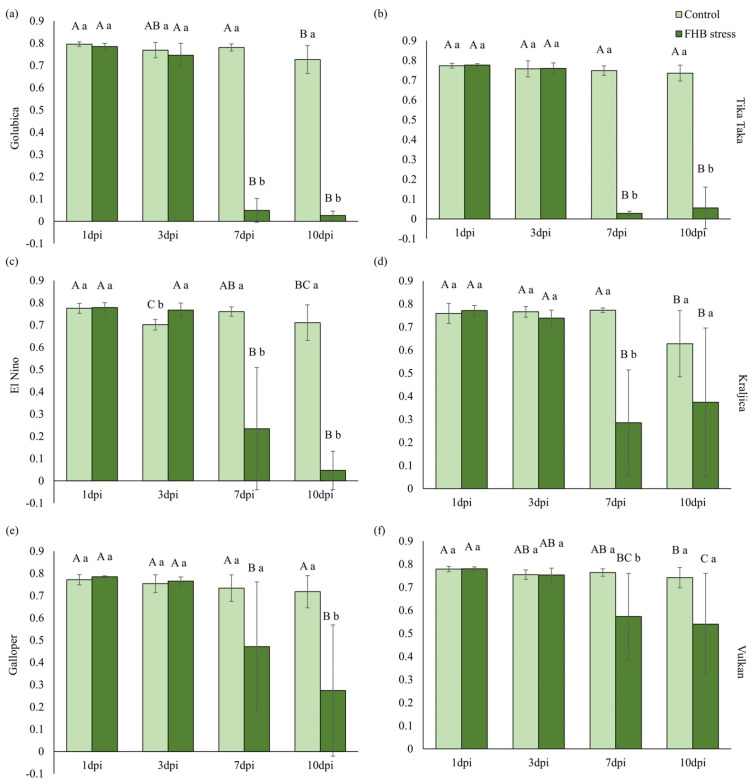
Maximum quantum yield of primary photochemistry (TR_0_/ABS) in control and FHB-stressed spikes of variety Golubica (**a**), Tika Taka (**b**), El Nino (**c**), Kraljica (**d**), Galloper (**e**), and Vulkan (**f**). Bars represent mean values of six independent biological replicates ± SD. Different small letters indicate significant difference between treatments at each measurement point separately (1 day post inoculation (dpi), 3 dpi, 7 dpi, and 10 dpi) (*p* < 0.05). Different capital letters indicate difference among measurement points in each treatment separately (*p* < 0.05).

**Figure 6 plants-12-03720-f006:**
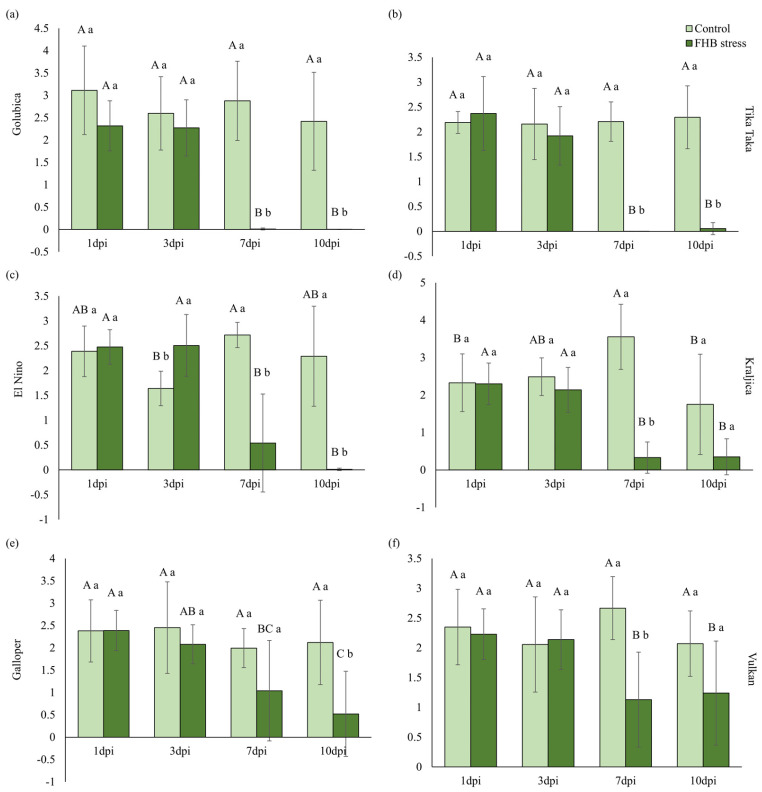
Performance index on absorption basis (PI_abs_) in control and FHB-stressed spikes of variety Golubica (**a**), Tika Taka (**b**), El Nino (**c**), Kraljica (**d**), Galloper (**e**), and Vulkan (**f**). Bars represent mean values of six independent biological replicates ± SD. Different small letters indicate significant difference between treatments at each measurement point separately (1 day post inoculation (dpi), 3 dpi, 7 dpi, and 10 dpi) (*p* < 0.05). Different capital letters indicate difference among measurement points in each treatment separately (*p* < 0.05).

**Figure 7 plants-12-03720-f007:**
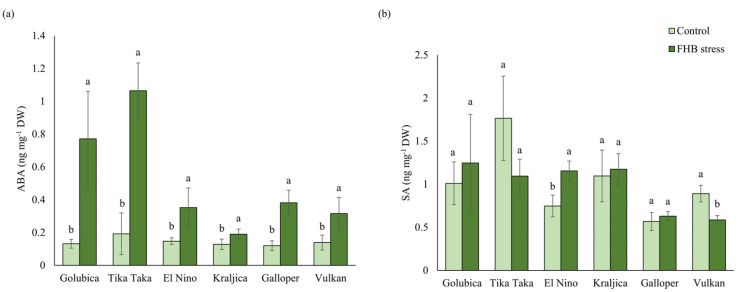
Content of (**a**) abscisic acid (ABA) and (**b**) salicylic acid (SA) in control and FHB-stressed spikes of six winter wheat varieties (Golubica, Tika Taka, El Nino, Kraljica, Galloper, and Vulkan). Bars represent mean values of four independent biological replicates ± SD. Different letters indicate significant difference among treatments in each variety separately (*p* < 0.05).

**Figure 8 plants-12-03720-f008:**
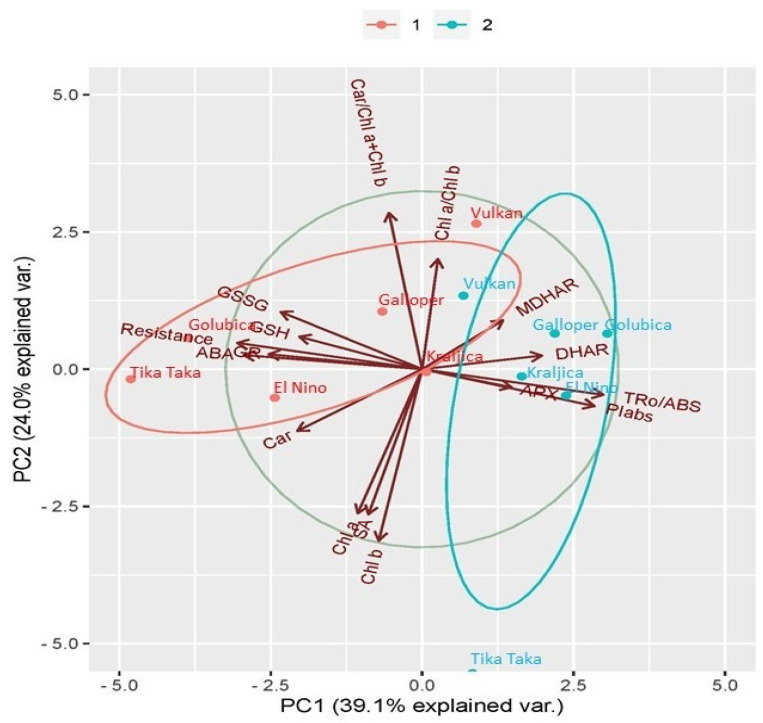
Principal component analysis (PCA) showing the relationship between reduced glutathione (GSH), oxidized glutathione (GSSG), ascorbate peroxidase (APX), monodehydroascorbate reductase (MDHAR), dehydroascorbate reductase (DHAR), glutathione reductase (GR), content of chlorophyll *a* (Chl *a*), chlorophyll *b* (Chl *b*), carotenoids (Car), chlorophyll *a*/*b* ratio (Chl *a*/Chl *b*), carotenoids/total chlorophyll ratio (Car/Chl *a* + Chl *b*), maximum quantum yield of primary photochemistry (TR_0_/ABS) measured at 10 days post inoculation (dpi), performance index on absorption basis (PI_abs_) measured at 10 dpi, abscisic acid (ABA), salicylic acid (SA), and type II resistance (Resistance) in the control (blue colored) and FHB-stressed (red colored) spikes of six winter wheat varieties (Golubica, Tika Taka, El Nino, Kraljica, Galloper, and Vulkan).

**Table 1 plants-12-03720-t001:** Values for type II resistance to Fusarium head blight (resistance to disease spread within the spike) of six winter wheat varieties. Data are average values of six independent biological replicates ± SD. Different letters indicate significant difference between varieties (*p* < 0.05).

Variety	Number of Infected Spikelets on 10 dpi
Golubica	3.7 ± 0.47 ^a^
Tika Taka	3.5 ± 0.76 ^ab^
El Nino	2.5 ± 0.96 ^bc^
Kraljica	2 ± 0.58 ^cd^
Galloper	1.8 ± 1.46 ^cd^
Vulkan	1 ± 0.82 ^d^

## Data Availability

All of the data are contained within the article. The raw MS files are available on request from the corresponding author.

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
