# Peer review of "Fusarium Head Blight Infection Induced Responses of Six Winter Wheat Varieties in Ascorbate–Glutathione Pathway, Photosynthetic Efficiency and Stress Hormones"

_plants, 2023, doi:10.3390/plants12213720_

Round 1

Reviewer 1 Report

Comments and Suggestions for Authors

Review of Manuscript „Fusarium Head Blight Infection Induced Responses of Six Winter Wheat Varieties in Ascorbate-Glutathione Pathway, Photosynthetic Efficiency and Stress Hormones”

 I find the research interesting and it contains elements of scientific novelty. The manuscript is generally well-written, but I believe that the authors should supplement the data analysis with some of the elements indicated below. Therefore, I recommend that the article be accepted for publication with major revision.

Here are my specific comments:

1. Table 1: Please perform statistical analysis of data.

 2. Please add photo material showing the different resistance of wheat varieties to FHB. 3. Figures 4 and 5: Please describe exactly what the labeling of the graph bars with small and large letters refers to. It is not legible.

4. I suggest that correlation analyses should be performed using Spearman's rank method, allowing for linkage of cultivar resistance to the traits studied (i.e., GSH, GSSG, APX, MDHAR, DHAR, GR, content of chlorophyll, and others). The analysis should be performed separately for control and F. culmorum and F. graminearum spore-infected objects.

5. Please answer how the time after inoculation influenced the parameters studied (Spearman rank correlation analysis between time after infection and the traits studied). The analyses should be performed separately for control plants and plants infected with F. culmorum and F. graminearum spores.

6. I believe that future research should be directed towards clarifying the effect of mycotoxins formed by Fusarium on the parameters studied by the authors. I hope this helps, and I look forward to seeing the revised manuscript.

Author Response

Dear Reviewer 1,

please find our answers to your comments and review in the attachment.

Reviewer 2 Report

Comments and Suggestions for Authors

The research evaluated the physiological and biochemical responses of winter wheat to FHB stress on the AsA-GSH pathway, photosynthetic efficiency and stress hormone levels.

The research is interesting given the fact that it is not common to evaluate pathogen damage in physiological mechanisms. There is a need to improve the quality of the manuscript. In the methodology it is important to include the name of the soil and chemical analysis of the soil. It seems that the authors applied NPK in the form of a nutrient solution but this needs to be made clear. The authors must indicate the amount of N, P and K applied per dm3 of soil so that you can understand what was applied in terms of quantity and also the source of the fertilizer used. Why didn't you apply micronutrients? How was irrigation management carried out? This has to be clear. As this is research involving a pathogen and we know that plant nutrition can affect the incidence of disease, these aspects must be clear in the manuscript.

It is important for the authors to detail the leaf sampled for the evaluations carried out and the part of the leaf blade (basal, median or apical) analyzed and whether the plant tissue damaged or not by the pathogen was chosen.

We strongly recommend that you carry out multivariate analysis to improve the discussion or defense of the results obtained and, therefore, the quality of the manuscript. It is important for authors to include the photosynthetic rate.

Author Response

Dear Reviewer 2,

please find our answers to your comments and review in the attachment.

Round 2

Reviewer 1 Report

Comments and Suggestions for Authors

There have been revisions to the manuscript. The authors have answered all of the queries. The paper can be printed in its existing format. I congratulate the authors on their interesting research.

Reviewer 2 Report

Comments and Suggestions for Authors

There was improvement in the manuscript with the revision of the manuscript.